# Streamable Portrait Video Editing with Probabilistic Pixel Correspondence

## ABSTRACT

Portrait video editing has attracted wide attention thanks to its practical applications. Existing methods either target fixed-length clips or perform temporally inconsistent per-frame editing. In this work, we present a brand new system, StreamEdit, which is primarily designed to edit streaming videos. Our system follows the ideology of **editing propagation** to ensure temporal consistency. Concretely, we choose to edit only one reference frame and warp the outcome to obtain the editing results of other frames. For this purpose, we employ a warping module, aided by a probabilistic pixel correspondence estimation network, to help establish the pixel-wise mapping between two frames. However, such a pipeline requires the reference frame to contain all contents appearing in the video, which is scarcely possible especially when there exist large motions and occlusions. To address this challenge, we propose to adaptively replace the reference frame, benefiting from a heuristic strategy referring to the overall pixel mapping uncertainty. That way, we can easily align the editing of the before- and after-replacement reference frames via image inpainting. Extensive experimental results demonstrate the effectiveness and generalizability of our approach in editing streaming portrait videos. Code will be made public.

## CCS CONCEPTS

• **Computing methodologies** → **Video editing; Generative Multimedia; Computer vision**;.

## KEYWORDS

portrait video processing, propagation-based video editing, diffusion model;

## 1 INTRODUCTION

Portrait video editing plays a critical role in enhancing aesthetics in content creation, bolstering viewer engagement in live streaming, and improving immersive experiences in virtual reality. The evolution of generative models[8, 12, 18, 24, 41] has markedly improved the performance of portrait editing, particularly in terms of fidelity. Nevertheless, portrait video editing remains a complex task due to the requirement for high precision in capturing and modifying subtle expressions and movements while maintaining excellent temporal consistency.

*ACM MM, 2024, Melbourne, Australia*
© 2024 Copyright held by the owner/author(s). Publication rights licensed to ACM.
ACM ISBN 978-x-xxxx-xxxx-x/YY/MM
https://doi.org/10.1145/nnnnnnn.nnnnnnn

**Unpublished working draft. Not for distribution.**

While numerous methods have been proposed to achieve consistent portrait editing, they often encounter various drawbacks. Talking head [19, 48, 52, 54, 59, 62] uses extensive human face priors and successfully enables highly consistent long video editing; however, its applicability is greatly limited as edits are constrained to the head only. Recent advancements in Text-to-Image diffusion models[8, 18, 41] have inspired a wave of zero-shot video editing techniques [11, 34, 49, 51] that incorporate temporal modules with cross attention. Despite these developments, they only manage to alleviate temporal flickering rather than eliminate it entirely due to the inherent randomness in the generation process. Reference-based [21, 22, 38, 50] and atlas-based [25, 33] propagation methods find it challenging to capture subtle movements, particularly in longer videos, a scenario in which the efficacy of both the reference frame and the atlas diminishes.

In this study, we propose a method for portrait video editing that incorporates probabilistic pixel correspondence. Specifically, we combine the strengths of landmark-based, propagation-based and large-model-based method by designing landmark warping modules that utilize pre-trained DINOv2 features. This approach allows us to capture small facial movements with the initialized landmarks, while other parts can be reconstructed leveraging the capabilities of DINOv2 features. As a result, our pipeline is not restricted to the head and can also handle body parts. Another challenge we address is the appearance of occluded regions, especially in longer videos, where pixels may not correspond to the reference image. To identify these non-corresponding pixels, we propose a probabilistic correspondence estimation network that takes the reference and current image as inputs and outputs dense correspondences and uncertainties for each pixel. Leveraging the learned uncertainty, we introduce an adaptive reference replacement scheme to dynamically update the reference image.

With our proposed pipeline, we can perform high-fidelity portrait editing while maintaining excellent temporal consistency. We carry out comprehensive experiments, which demonstrate that our method surpasses all baseline measures in video reconstruction quantitatively, and it also significantly outperforms in user studies. Keys to our approach are the proposed modules: probabilistic pixel correspondence estimation and adaptive reference replacement. The former module effectively captures the fine details of movement, while the latter adjusts the reference to accommodate occluded content. We conduct ablation studies for these modules, demonstrating their efficacy. Furthermore, we illustrate that our model, once trained on the initial frames, can seamlessly transfer to subsequent incoming frames. This design highlights our solution's potential for streaming applications.

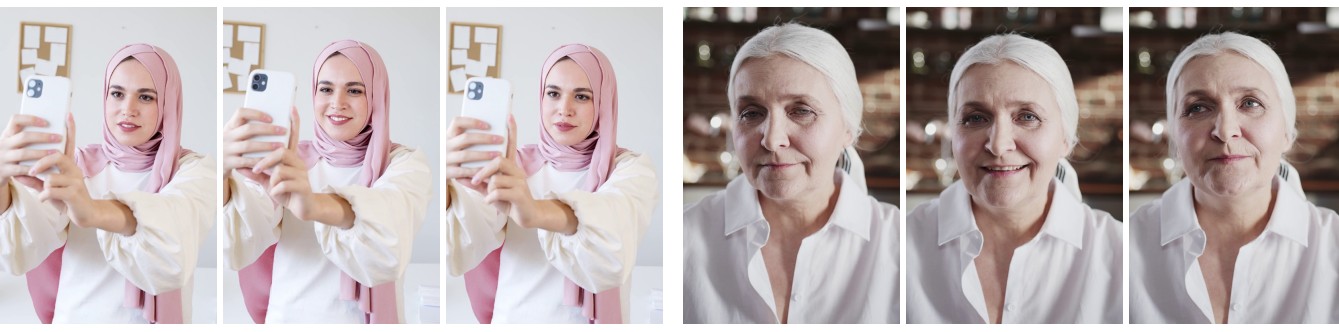

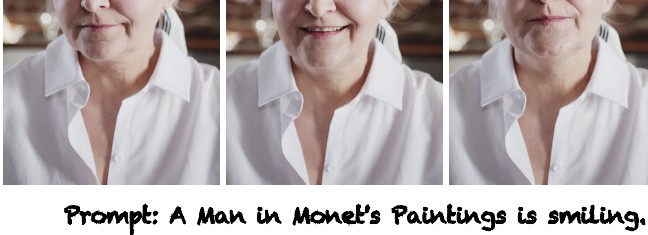

Prompt: A Zelda girl is taking a selfie.

Prompt: A Man in Monet's Paintings is smiling.

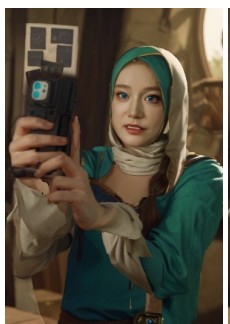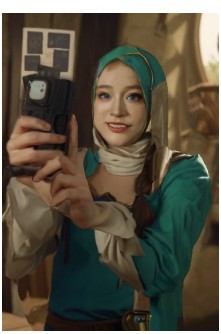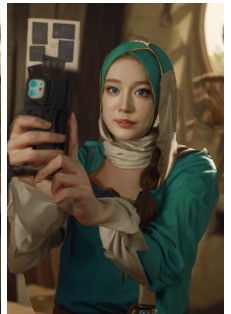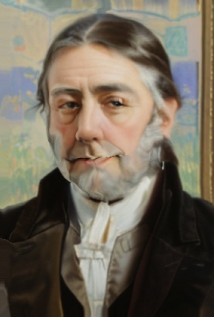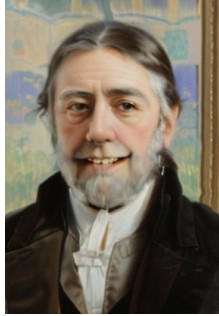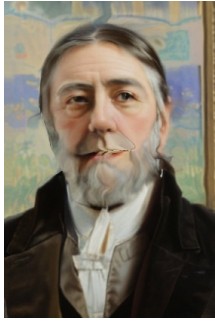

**Figure 1: Photo-realistic portrait video editing results. Our approach to photo-realistic portrait video editing yields impressive results, which has a unique capability to effectively handle motions and occlusion from portrait videos. Consequently, it generates images that are high in fidelity and maintain temporal consistency, ensuring the continuity of the narrative throughout the video sequence.**

## 2 RELATED WORK

**Portrait video processing.** Portrait video processing has attracted considerable attention due to its profound practical applications. This field encompasses a range of tasks, from human video editing [22, 25, 28, 29, 33, 43], human style transfer [9, 10, 23, 42] to talking head video generation [15, 19, 40, 46, 48, 52, 54, 57–59, 62, 63], all of which extensively leverage human priors like facial landmarks.

Yao *et al.* [55] and Tzaban *et al.* [44] employ facial landmarks to align and crop the target face area for facial editing in real videos. However, their approaches only incorporate facial priors in the pre-processing stage, leading to inconsistent results. Furthermore, Kim *et al.* [26] applies a pretrained landmark detection model to extract per-frame motion information. Following this process, they introduces a landmark encoder to ensure temporal consistency in face video editing In addition to facial landmark priors, 3D morphable priors are also used in video editing. Cao *et al.* [4] integrate a 3DMM reconstruction module, designed to decompose a video portrait into pose, expression, and identity coefficients.

While the methods mentioned above focus on real face video editing, the study of human stylization is also a well-established field with broad applications in many areas. DST [27] is the pioneering method that integrates geometry priors, such as face keypoints, into one-shot, domain-agnostic style transfer, yielding remarkable outcomes. Cui *et al.* [6] introduce a method for one-shot stylization of full-body human images very recently.

In talking head video synthesis, a number of studies [15, 58, 59, 63] utilize facial landmarks, identified by an off-the-shelf face model [14], as the anchor points of the face. Following this, the facial motion flow, derived from these landmarks, is transferred from a driving face video. Nevertheless, the motion flow in these studies is susceptible to cumulative errors due to the inaccuracies inherent in the face model. To circumvent this limitation, some other works [19, 40, 48, 62] resort to unsupervised learning, which provides a more accurate representation of facial motion by incorporating improved mechanisms that model the motion transformation between two sets of keypoints. While these methods are constrained to using facial landmarks as priors, leading to difficulties in dealing with other body parts like hands. Our approach goes beyond integrating facial landmark priors; it also employs DINOv2 [32] features and a cross-attention [45] mechanism for precise alignment of body parts. The pretrained DINOv2 features can also adapt to unseen upcoming frames, endowing our method with the capacity for streamability.

**Propagation-based consistent video editing.** Consistent video editing [20–22, 25, 28, 29, 33, 38, 43, 50] is a longstanding problem in computer vision field, and our work is closely intertwined with addressing this challenge. We mainly discuss the propagation-based techniques [20–22, 38, 43, 50], which involve initially editing a keyframe and then propagating these edits throughout all the video frames. Although this kind of approach is simple and computationally efficient, it may lead to inaccuracies and inconsistencies

**Figure 2: Pipeline Overview. (a) The introduction of Adaptive Reference Replacement (ARR), a module that utilizes an uncertainty map to dynamically determine whether to update a reference frame, aims to handle self-occlusion cases, such as hair, as depicted in the illustration above.(b) Probabilistic Pixel Correspondence Estimation (PPCE) utilizes DINOV2 features and face landmark information to estimate the displacement of the reference frame and uncertainty map.(c) The process of propagating the edited results involves utilizing the uncertainty map to guide the inpainting process, ensuring that only pixels with a high likelihood of error are modified.**

when edits are propagated across the temporal dimension. Moreover, when applied to human-centric videos, these methods often yield inferior results due to their lack of specifically tailored designs for human subjects. A primary issue with these methods is their reliance on a single reference frame, which makes handling motions and occlusions a challenge. A viable solution is to "envision" the occluded area. With the progress in generative models like GANs [12, 24] and diffusion models[8, 18, 41], image inpainting [1, 7, 30, 39, 56] has demonstrated potential in creating contextually fitting and plausible content. In our work, we resort to the image inpainting technique. More specifically, we perform inpainting guided by a learned uncertainty map. This approach allows us to inpaint the smallest possible set of pixels, thereby ensuring maximum consistency.

**Video editing via large generative models.** The advances in diffusion models [8, 18, 41] have remarkably improve the generated outputs in text-to-image (T2I) tasks. Cutting-edge T2I diffusion models, including DALL-E series [2, 35, 36], Imagen [17], and Stable Diffusion [37], possess billions of parameters and have been trained on extensive images. As such, they boast exceptional generative capabilities.

Building upon these T2I models, numerous derivative models [3, 31, 47, 60] have emerged, incorporating additional conditions such as depth maps, edge maps, and normal maps to enhance the controllability of the generation process. Based on these works, T2I-based video editing is gaining increasing popularity. Approaches such as Tune-A-Video [51], FateZero [34], Vid2Vid-Zero [49] and TokenFlow [11] delve into the latent space of diffusion models and strive for feature space matching across frames. For instance, they establish cross-frame attention maps to enhance consistency in video editing. Despite their advancements, these methods have not yet fundamentally addressed the issue of consistency especially for long videos, largely due to the manipulation solely within the features. Instead, we employ a hybrid approach that combines facial landmarks with DINOv2 features and achieve superior consistency and quality.

## 3 METHOD

Our framework follows the ideology of *editing propagation*, which entails first modifying a keyframe as the reference image and then disseminating these alterations across all the video frames. The consistency of the video is ensured by constraining the video's

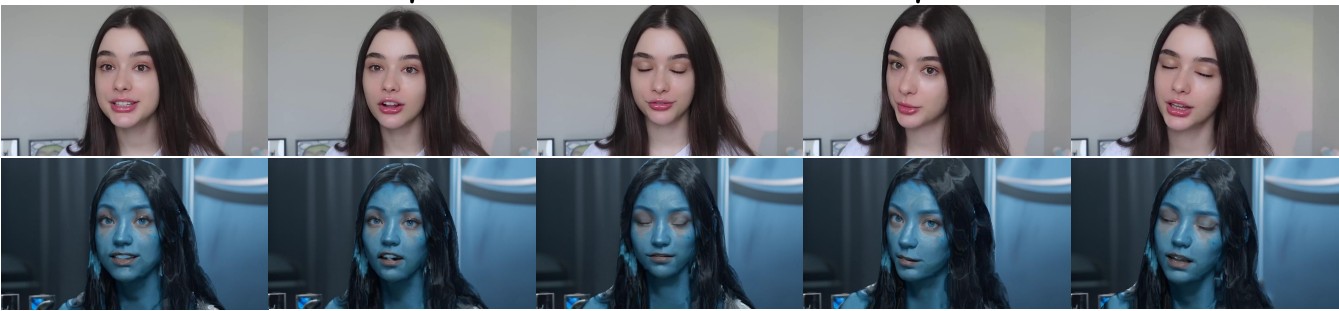

**Prompt: Van Gogh, a Man is Holding a Microphone**

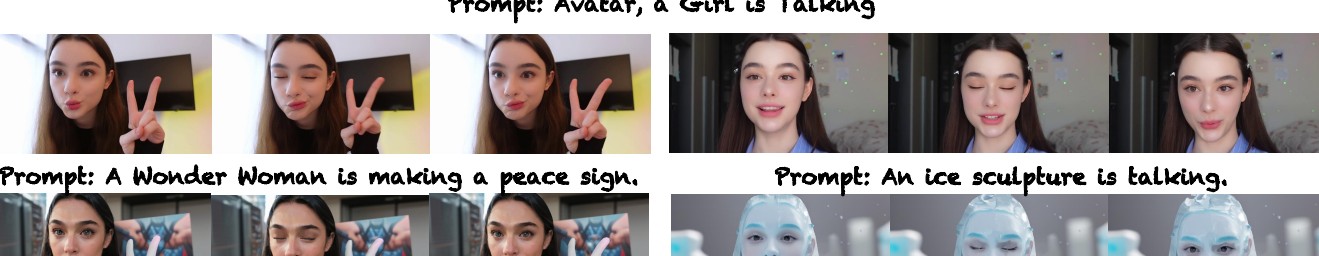

**Prompt: Avatar, a Girl is Talking**

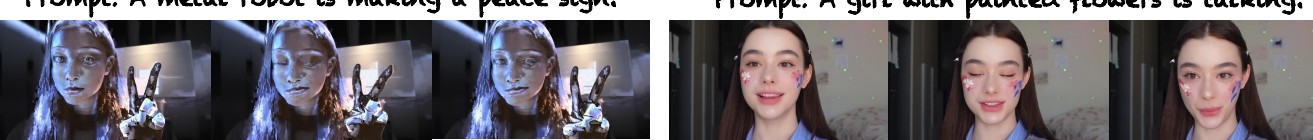

**Prompt: A Wonder Woman is making a peace sign.** **Prompt: An ice sculpture is talking.**

**Prompt: A metal robot is making a peace sign.** **Prompt: A girl with painted flowers is talking.**

**Figure 3: More qualitative results. Our approach showcases its effectiveness in not just modifying head and facial movements but also effectively managing simple hand gestures. Additionally, we can also edit facial details, such as painting flowers on the face, while maintaining facial structural stability.**

appearance to be sampled from a single 2D image. This methodology is essentially constructed on a substantial premise, which assumes that pixels in different frames corresponding to the same points should maintain identical colors. However, in real-world scenarios, this assumption does not hold true. In cases of occlusion or large motions, pixels cannot consistently locate their legitimate corresponding points in the reference image. Consequently, these occluded regions may lead to inconsistencies. This limitation also implies that such an approach can only be applied to videos of a fixed length. To generalize the editing propagation process so that

it can handle longer videos featuring extensive motion and occlusion, our `StreamEdit` incorporates the following submodules, as depicted in fig. 2: *Dense Probabilistic Pixel Corresponding* (section 3.1), *Adaptive Reference Replacement* (section 3.2) and *Uncertainty-driven Inpainting* (section 3.3) for streamable editing.

## 3.1 Probabilistic Pixel Correspondence

The key of the editing propagation is to find the dense correspondence between the current image $I_{curr}$ and the reference image $I_{ref}$. Our goal is to learn a function $F_{warp} : (u, v) \rightarrow (u', v')$, where $(u, v) \in I_{curr}, (u', v') \in I_{ref}$.

**Dino-Landmarks guided dense correspondence.** In order to handle complex motions and occlusions associated with human objects in video sequences, we propose to employ the face landmarks to guide the dense warping $F_{warp}$. As shown in fig. 2, we first extract the facial landmarks from each frame in the video. We then calculate the barycentric interpolation to obtain a dense warp field $F_{Lwarp}$ on the face. This process enables us to obtain a partially warped image via landmarks interpolation: $I_{Lwarp} = F_{Lwarp}(I_{curr})$. While facial landmarks ensure smooth and dense interpolation on the face region, it is challenging to generalize to other parts of the human portrait. Hence, to harmonize the warp field $F_{Lwarp}$ with other parts of the portrait, we leverage the pre-trained feature encoder, DINOv2 [32], to extrapolate the correspondence from the facial region. By leveraging the highly robust DINOV2 feature, we not only achieve precise alignment of the face and body but also effectively handle the matching of regions with intricate texture details, including hair and teeth. The advanced capabilities of DINOV2 enable us to perform accurate warping and alignment, even in areas with complex textures, ensuring a seamless and natural result.

Specifically, the features of the reference image, warped image and current image extracted by DINOv2 are denoted as $F_{ref}$, $F_{warp}$ and $F_{curr}$ respectively. Furthermore, we introduce a novel cross-attention mechanism to enhance the image features and produce comprehensive and dense correspondences:

$$I_{warp}, I_{unc} = CrossAttn(F_{curr}, F_{ref}, F'_{warp}), \qquad (1)$$

where $I_{warp}$ is the warped image from reference image and $I_{unc}$ is the uncertainty map.

**Uncertainty modeling.** For the image propagation, the photometric consistency assumption may be violated by the illumination changes and occlusions. However, we can assume the the photometric residuals of $|I_{ref} - I_{curr}|$ will satisfy the laplace distribution when only noises are added in this system. Inspired by the previous methods which assume the noise as a laplacian distribution [53], the negative log-likelihood to be minimized is

$$-\log p(y|\hat{y}, \sigma) = \frac{|y - \hat{y}|}{\sigma} + \log \sigma + C,$$
$$\mathcal{L}_{unc} = \frac{|y - \hat{y}|}{\sigma} + \log \sigma + C \qquad (2)$$

where $C$ is a constant and $\sigma$ is the uncertainty predicted from $I_{unc}$.

## 3.2 Adaptive Replacement of Reference Image

The uncertainty module aids in determining whether pixels can locate their correspondence in the reference image. For longer videos, occluded areas and unseen scenes can create inconsistencies. This issue confines previous methods to operating only within limited video lengths. To overcome this, we propose an adaptive strategy for replacing the reference image, which allows for dynamic changes to the reference image. For a long video $V = \{I_1, I_2, ..., I_n\}$, we take the first image $I_1$ as the initial reference image $I_{ref}$, and construct the first window $S = I_1, ..., I_{S+1}$, here $S$ is the maximum length of the current sliding windows. And the last frame $I_{S+1}$ in the window find its correspondences to the current reference image $I_{ref}$ with the probabilistic pixels correspondence module above $I_{unc}, I_{warped} = F_{warp}(I_{ref}, I_{S+1})$. If the uncertainty map $I_{unc}$ is above

a certain threshold, we will decide to split the video into a new sliding window and a new reference image $I_{ref'}$. This adaptive replacement technique ensures effective handling of scenarios where the reference frame is insufficient to cover the entire video content. It also ensures the preservation of pixel correspondence for the majority of frames. By selectively updating only a small number of error pixels, it guarantees excellent continuity, thereby maintaining the overall coherence of the video.

## 3.3 Uncertainty-driven Inpainting for Editing

After learning the dense warping and uncertainty, the video editing can be consistently propagate within the local windows as shown in fig. 2. Firstly, we edited the first reference image by the text guided image editing algorithms such as ControlNet [60]: $I^0_{edit} = \Phi(I^0_{ref}, text)$, and we can propagate the result using the wrap result during reconstrcution process. Using the dense correspondence between the current image $I_{curr}$ and the reference image $I_{ref}$. Leavaging the function $F_{warp}$. We can propagate the editing result by $I^{curr}_{edit} = F_{warp}(I^0_{edit})$. And for the sliding windows split in the section 3.2, the new appending reference image is first get the warping by the learnt probabilistic corresponding module, and we mask those pixels and inpaint them with the existed inpainting networks.

## 3.4 Training Losses

To supervise the correspondence between the queried images and the multiple reference images. The overall training objective is expressed as follows:

$$\mathcal{L}_{total} = \alpha_1 \mathcal{L}_{rec} + \alpha_2 \mathcal{L}_{unc},$$
$$\mathcal{L}_{rec} = \sum_{i=1, I_j \in \Omega_i}^{M} ||I_j - \mathcal{W}(I_j, I^i_{ref})||, \qquad (3)$$

where $\alpha_1, \alpha_2$ are two hyper parameters and $\Omega_i$ is the images set in the adaptive replacement of $I^i_{ref}$, $\mathcal{W}$ is the warping operation.

## 4 EXPERIMENTS

### 4.1 Experimental Setup

In order to demonstrate the superior performance and robustness of our method, we conducted comprehensive experiments. Firstly, we evaluated the stability and temporal consistency of our method on long videos through a rigorous analysis on the HDTF [61] dataset, which comprises 57 high-resolution human videos exceeding one minute in duration. Furthermore, to demonstrate the practical applicability of our method, we extensively tested our text-based edited results on a diverse set of web videos featuring complex scenes, encompassing a wider range of expressions and more pronounced body movements. We also demonstrate that with the trained models, we can apply edits to the newly incoming frames in streaming videos. Finally, we conducted an ablation study to demonstrate the effectiveness of the modules designed in our approach.

For the specific implementation details, we utilized Segment-Anything-track (SAMtrack) [5] to extract segmentation results of foreground figures from the video. Additionally, we employed MediaPipe [13] to extract face landmarks. The training process was conducted with a maximum of 40,000 iterative steps, with periodic

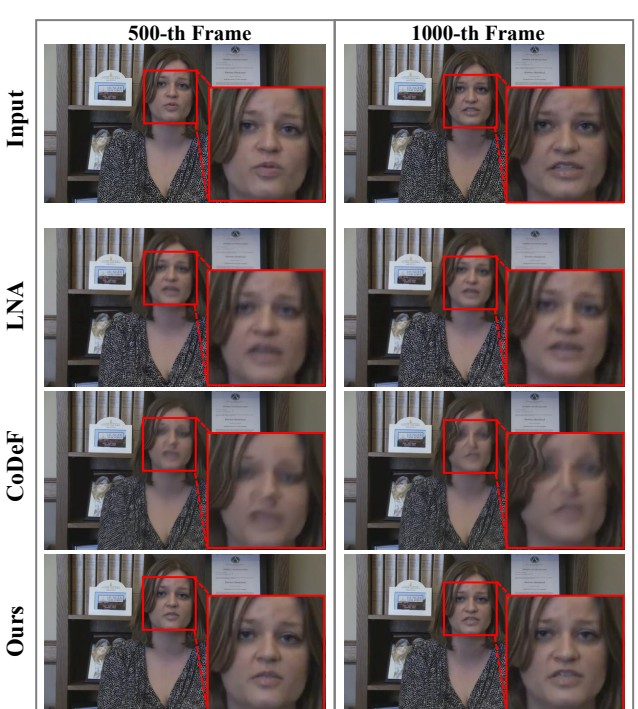

Figure 4: Reconstruction comparison on HDTF [61] dataset. We compare our method with Layered Neural Atlas (LN-Atlas) [25] and CoDeF [33], demonstrating the superiority of our method.

evaluations every 10,000 iterations to assess whether the uncertainty of frames exceeded the predetermined threshold. On a single NVIDIA 3090 GPU, the average training time for a video comprising 150 frames, each with a size of 768x432 pixels, was approximately 45 minutes. Regarding inference time, it can achieve a speed of nearly 5 frames per second.

## 4.2 Reconstruction Quality on HDTF

For reconstruction testing, we sampled a thousand consecutive frames from each video in the HDTF [61] test set. To evaluate the quality of the reconstructions, we employed three widely used metrics: structural similarity index (SSIM), peak signal-to-noise ratio (PSNR), and perceptual image similarity (LPIPS). In table 1, we present a comparison of our StreamEdit method with state-of-the-art layered representation-based approaches, namely Layered Neural Atlas [25] and CoDeF [33]. It can be observed that our StreamEdit obtains the best results compared with other competitive methods. Besides, as illustrated in fig. 4, our approach excels in effectively preserving detailed facial movements in long videos.

Table 1: Reconstruction result on the HDTF [61] dataset.

| Models | SSIM (%) ↑ | PSNR ↑ | LPIPS ↓ |
|---|---|---|---|
| Layered Neural Atlas [25] | 94.7 | 29.63 | 0.072 |
| CoDeF [33] | 95.2 | 30.51 | 0.066 |
| StreamEdit | **97.4** | **33.41** | **0.027** |

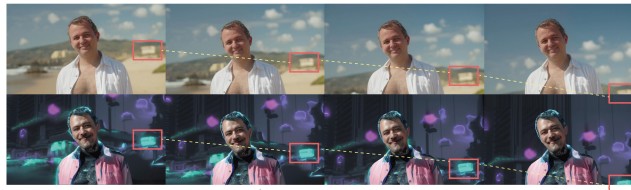

Prompt: Cyberpunk, a Man is Smiling

Figure 5: Editing Results with Background Motion. Our method learns complex pixel-level matching relationships for layered foreground and background elements, thus enabling users to edit videos with camera motion.

## 4.3 Editing Results on Diverse Web Videos

To further assess the effectiveness and practicality of our approach, we conducted extensive evaluations using rich web videos that encompassed a broader range of micro-expressions and head movements. We employed both objective metrics and human preferences to evaluate the performance.

For text fidelity evaluation, we utilized the average CLIPScore [16] between the output video frames and their corresponding text descriptions as a measure. This metric quantifies the degree of fidelity between the generated visual content and the intended textual representation. Moreover, to gauge human preference, we enlisted the participation of 77 volunteers. We presented them with the baseline editing results and the corresponding text descriptions and requested them to score ranging from 1 to 5 on four dimensions: Motion Coherence (MC), Text Fidelity (TF), Temporal Consistency (TC), and Overall Quality. The average score across the volunteers was then computed to derive the final result for each evaluation metric.

We conducted both quantitative and qualitative comparisons of our method with Tune-A-Video [51], FateZero [34], CoDeF [33] and TokenFlow [11]. The results presented in fig. 6 provide compelling evidence that our method excels in maintaining consistency while editing details, such as teeth and hair. It is essential to preserve intricate facial movements as they play a pivotal role in effectively conveying emotions. However, baseline methods fail to preserve complex facial movements, such as eye closure and mouth movements, resulting in poor editing results. Tune-A-Video [51] faces challenges in ensuring consistent positioning of the characters relative to the original video, resulting in significant spatial displacement. Insufficient understanding of the semantics of the learned canonical images in CoDeF [33] results in distorted facial expressions. In the case of Tokenflow [11], its propagation in an implicit space limits its ability to capture subtle movements and results in similar editing.

Additionally, the user study in table 2 shows that our approach outperforms the others in terms of visual results, particularly in maintaining temporal consistency. As evident in table 2, our method achieves the highest human preference in all aspects and outperforms all baselines by a large margin.

Furthermore, our approach successfully learns pixel correspondence for both portrait regions and background regions, as illustrated in fig. 2. This capability enables natural background editing, such as adjustments caused by camera motion. Notably, our method

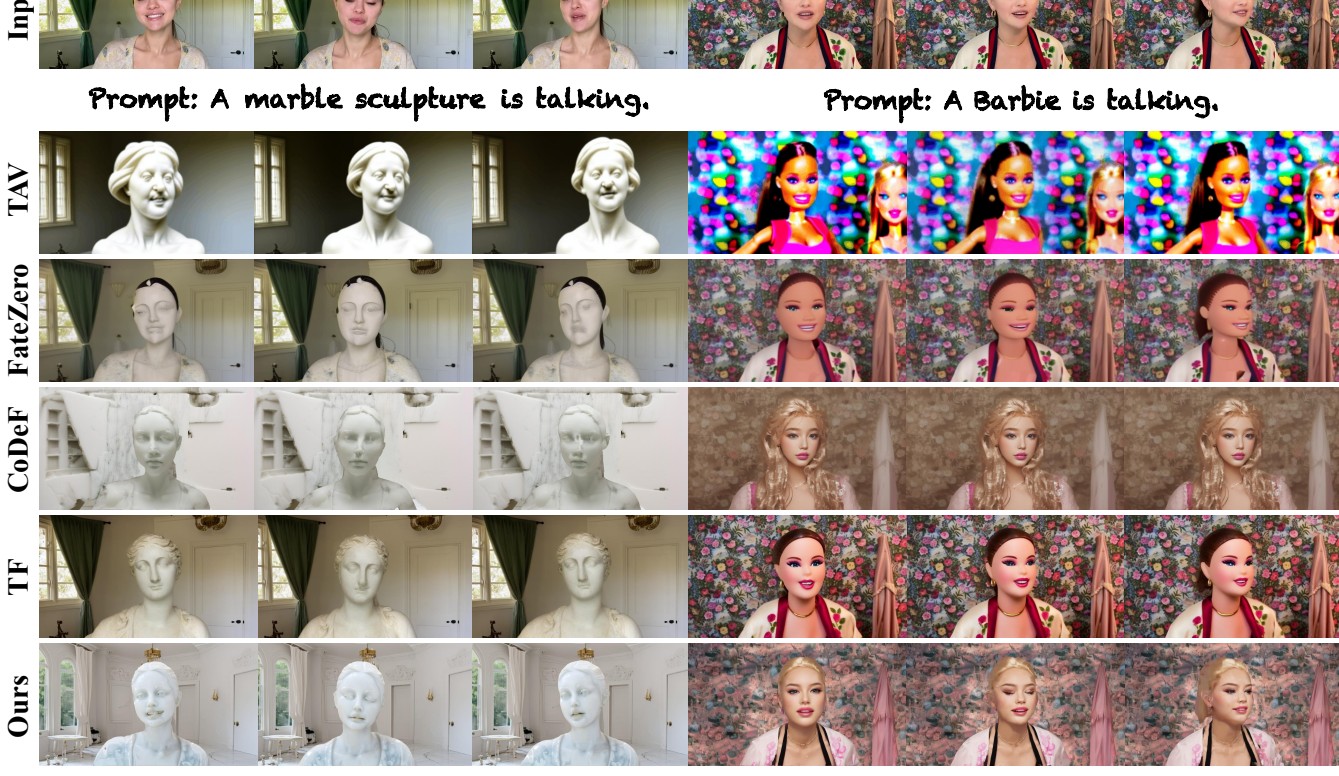

**Figure 6: Visual results on Diverse Web Videos. We compare our method against Tune-A-Video(TAV) [51],FateZero [34], CoDeF [33], TokenFlow(TF) [11].**

Table 2: Quantitative results on Diverse Web Videos.

| Model | CLIPScore ↑ | MC ↑ | TF ↑ | TC↑ | Overall ↑ |
|---|---|---|---|---|---|
| Tune-A-Video [51] | 27.62 | 3.13 | 3.29 | 3.02 | 3.00 |
| FateZero [34] | 28.13 | 3.71 | 3.67 | 3.4 | 3.21 |
| CoDeF [33] | **28.87** | 3.41 | 3.50 | 3.27 | 3.29 |
| TokenFlow [11] | 28.57 | 3.62 | 3.73 | 3.59 | 3.52 |
| StreamEdit (ours) | 27.87 | **4.53** | **4.25** | **4.48** | **4.36** |

effectively handles background motion, resulting in consistent and high-quality edits, as demonstrated in the fig. 5.

## 4.4 Streamability Demonstration

We evaluate the streamability of our pipeline. Our model is optimized on the initial 1000 frames and the editing outcomes are tested on the subsequent 500 frames. As depicted in fig. 7, our model exhibits the ability to generalize to new frames, provided that the reference has been updated.

Moreover, we conducted speed tests to evaluate the efficiency of our editing process. Once the training of the reconstruction is completed, our method achieves an impressive editing processing speed of 10 frames per second. In comparison, Tune-A-Video[51] operates at a significantly slower speed of 0.6 fps, while TokenFlow[11] performs even lower at 0.3 fps. These results clearly indicate the superior efficiency of our method in terms of editing speed.

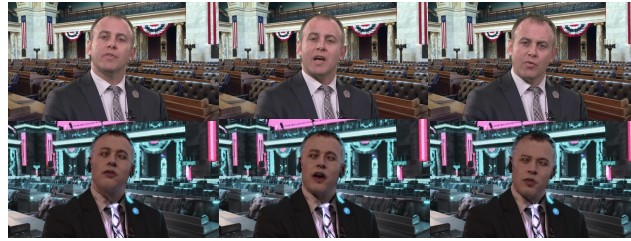

**Figure 7: Streamability demonstration. Once trained on the initial frames, our pipeline can seamlessly generalize to newly incoming frames *without* the need for further fine-tuning.**

## 4.5 Ablation Studies

In this section, we perform ablation studies to demonstrate the effectiveness of the proposed Probabilistic Pixel Correspondence Estimation (PPCE) and Adaptive Reference Replacement (ARR). The PPCE module plays a crucial role in modeling pixel-to-pixel correlations, enhancing the overall performance of our method. On the other hand, the ARR module proves to be highly effective in handling emerging objects and addressing challenges arising from self-occlusion scenarios.

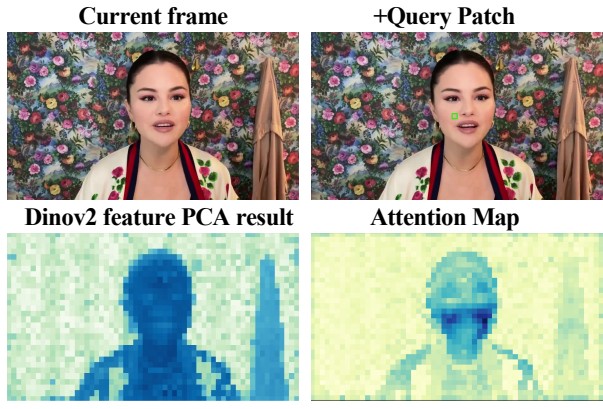

**Current frame**  **+Query Patch**

**Dinov2 feature PCA result**  **Attention Map**

**Figure 8: Visualization of Intermediate Results. From the PCA results, it can also be observed that the features of DINOV2 are highly robust. Besides, it is apparent that the query patch on the face effectively captures relevant information corresponding to the respective region in the reference frame.**

**Effect of probabilistic pixel correspondence estimation.** In fig. 8, we present the visualized cross-attention map of Landmarks-Attention module, revealing that the query patch on the face effectively captures information related to the corresponding region in the reference frame. This demonstrates that our designed Landmarks-Attention module further refines the DINOv2 [32] feature and enhances its applicability for matching. Additionally, by utilizing the mapping of landmarks between the reference and current video frames as the initial value, we simplify the process of fitting eye motion. The quantitative results in table 3 and the qualitative results in fig. 9 demonstrate that the absence of Landmarks-Guided warp module module leads to a noticeable decrease in PSNR due to the inability to accurately model closed-eye actions.

**Table 3: Ablation studies.**

| PPCE | ARR | SSIM (%) ↑ | PSNR ↑ | LPIPS ↓ |
|------|-----|------------|--------|---------|
| ✓ |   | 98.1 | 31.28 | 0.024 |
|   | ✓ | 98.0 | 31.11 | 0.021 |
| ✓ | ✓ | **99.1** | **34.58** | **0.014** |

**Effect of adaptive reference replacement.** The ARR module is essential for effectively dealing with occlusions and updating the reference frame. Removing ARR would result in the absence of hair when the head is turned sideways, as demonstrated in fig. 9. Furthermore, the quantitative results presented in table 3 provide additional evidence of the module's effectiveness.

## 5 CONCLUSION AND DISCUSSION

In this research, we introduce a novel approach to portrait video editing that leverages the integration of landmark warping and DINOv2 [32] features, facilitated by the utilization of probabilistic pixel correspondence. By employing this method, we achieve high-fidelity edits that retain temporal consistency, making them particularly well-suited for streaming scenarios. This innovative

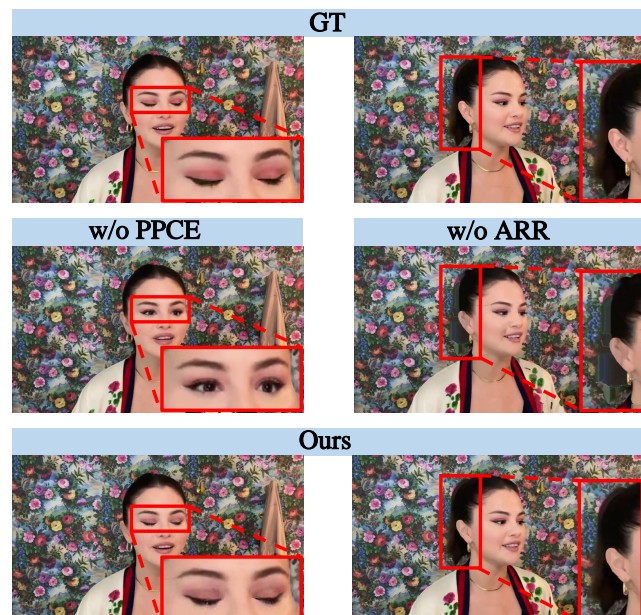

**GT**

**w/o PPCE**  **w/o ARR**

**Ours**

**Figure 9: Qualitative Ablation Study. The PPCE and ARR modules are necessary for accurate modeling of closed-eye actions and handling occluded object parts, respectively.**

combination of techniques allows for precise and seamless modifications throughout the video, resulting in visually appealing and coherent results.

Our StreamEdit offers a new perspective on tackling the task of long portrait video editing, where the editing is performed only once on the first video frame and then propagated to the subsequent frames with learned pixel correspondence. Besides the novel pipeline, the key challenges lie in learning accurate per-pixel correspondence and adequately replacing the reference frame to constantly adapt to the unseen video stream.

Our design enjoys three advantages: (1) temporally consistent portrait video editing with large motions, (2) customization of editing on both the portrait and the background, (3) editing beyond a fixed video clip by taking streamability into account.

Nonetheless, we encounter several challenges. The first limitation lies in the slow speed of DINOv2 feature extraction, which can significantly impact the real-time editing performance. This limitation restricts the system's responsiveness and may not be suitable for scenarios that require fast editing, such as interactive applications or live events. The second limitation arises from the bais of ControlNet[60] towards generating individuals with their eyes glued to the screen. This bias can make it challenging to achieve accurate eye alignment, leading to a mismatch of eyes in the editing results. This limitation adversely affects the visual realism and quality of the rendered individuals.

Despite these issues, our results showcase the potential for high-speed, temporally coherent portrait editing. We anticipate that future efforts will focus on advancing this framework, aiming to enhance its generalization capability while further accelerating its performance.

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
