# OpenReview forum: "Streamable Portrait Video Editing with Probabilistic Pixel Correspondence"
_acmmm.org/ACMMM/2024/Conference — MM2024 Poster_

### Official Review · Reviewer_kQFC · 2024-05-12

**Rating:** 4
**Confidence:** 3

**Summary:**

This paper proposes a method for portrait video editing that incorporates probabilistic pixel correspondence. It achieves good video editing effects through landmark warping modules that utilize pre-trained DINOv2 features and an adaptive reference replacement scheme.

**Strengths:**

1. The visual results presented in paper and supplementary materials appear to be of good quality.
2. The 'PPCE' proposed in the article is interesting and effective, and it offers good reference value.
3. The structure of the article is clear and progresses logically.

**Limitations:**

1. There is a spelling error in Figure 2, "Displacment Map." The figure of the pipeline, especially part (c), needs improvement.
2. Regarding the 'Adaptive Replacement of Reference Image' step, theoretically, the information source for the new reference image is still the first frame, which still fails to receive suitable information about 'occluded areas and unseen scenes'. If the purpose is merely to smooth the transition, what advantages does this method have compared to using the (i-1)th frame to predict the ith frame? It would be helpful to supplement this with corresponding ablation experiments and data. (If this issue cannot be resolved, then the innovation aspect of the article's pipeline is somewhat weak.)
3. The video in the supplementary materials needs to be consistent with the cases mentioned in the main text.

**Suitability:**

3

---

### Official Review · Reviewer_MSdi · 2024-05-15

**Rating:** 5
**Confidence:** 3

**Summary:**

The paper proposes a novel approach to portrait video edting that leverages the integration of landmark warping/propagation and DINOv2 features together to generate probabilistic pixel correspondences. The evaluation shows that the proposed method achieves a high temporal consistence and fidelity.

**Strengths:**

- Paper is well written and has a good structure with high quality figures.
- Related work section is quite comprehensive and contains the relevant work in the field of portrait video processing and adjacent fields.
- The proposed method is sound and has sufficient novelty.
- The evaluation is very extensive, covering both quantitative and subjective quality tests / comparisons.
  Furthermore, or the subjective tests, a large user group (77 volunteers) has been gathered.
- Evaluation shows that the proposed method achieves better quality than the state of the art in the majority of the measures.

**Limitations:**

- As a key component of the proposed algorithm is the extraction of facial landmarks, I am wondering why the authors used the relatively old facial landmark extraction method from Mediapipe (reference [13]), which is not state of the art anymore. There is no reason given in the paper why the authors chose this specific method instead of newer ones like the method by Wood et al ( https://arxiv.org/pdf/2204.02776).

**Suitability:**

2

---

### Official Review · Reviewer_QhNk · 2024-05-24

**Rating:** 3
**Confidence:** 3

**Summary:**

This paper proposes a new method to edit portrait videos using editing propagation. The probabilistic pixel correspondences are estimated by facial landmark warping and extrapolation using DINOv2 features. The reference image is adaptively updated according to the overall pixel uncertainty. The model trained on a fixed-size clip can be applied to subsequent frames. The model was tested and compared on a diverse set of web videos.

**Strengths:**

This method shows superior reconstruction quality compared to Layered Neural Atlas and CoDeF.

The method is straightforward and the provided video results are of high quality.

The user study shows its strength in temporal consistency, motion coherence, and text fidelity.

**Limitations:**

Technical concerns:

Why not use optical flow to estimate dense correspondence or at least use it to initialize the correspondence?

Why can the sparse landmark correspondence be propagated to the background?

What's the structure of the network described in Eq. 1? Is it a single cross-attention layer?

What's the uncertainty threshold used to determine the update of the reference image? Did all videos share the same parameter? How can we determine the best parameter? Maybe it is also related to the selection of $\sigma$ in Eq. 2.

What's the parameter setting of layered neural atlas and CoDeF? For example, how many layers were optimized?

Which inpainting method was adopted? What is the common proportion of missing pixels?

How does the number of training frames affect the editing result?

Evaluation:

It is hard to compare different results since their appearances are quite different. It would be better to show the results of several methods propagating the same initial frame. Besides, I am not sure whether the user ratings are appropriate. For example, directly comparing the temporal coherence between the new method (based on pixel-wise propagation) and TokenFlow is not that reasonable since TokenFlow can partly handle the illumination change, which will somewhat deteriorate temporal coherence.

What's the definition of "large motions"? I don't think the provided video clips have large motion. Showing edited videos with small, medium, and large motions separately would be better.

Similar to CoDeF, this method may propagate other image edits other than text-based ones. Did the authors try any other operators?

Paper writing:

Eq. 2: the symbols $y$ and $\hat{y}$ are not defined.

Eq. 3: the summation operation should be $\sum_{i=1}^{M} \sum_{I_j \in \Omega_i}$.

missing spaces in lines 43, 71, 516, 750, and 751.

line 917: bais $\to$ bias.

**Suitability:**

3

---

### Meta-Review · Area_Chair_27m6 · 2024-07-05

**Recommendation:** Accept (Poster)
**Confidence:** 4

**Metareview:**

Reviewers agree that the paper is well-written and well presented, comprehensively covering the related works. The proposed method has sufficient novelty and presents extensive evaluation including subjective tests conducted with 77 volunteers. The rebuttal addresses the concerns regarding optical-flow-based methods but does not effectively address the other concerns raised by the reviewers. Nonetheless all reviewers are learning towards an accept. Hence, considering the overall reviews and the author response I am recommending the paper for an accept.